

# Electrical Formation Factor of Clean Sand from Laboratory Measurements and Digital Rock Physics.

**By**

**Mohammed Ali Garba[1,2], Stephanie Vialle[2], Mahyar Madadi[2], Boris Gurevich[2]**

**and Maxim Lebedev[2].**

**[1]Department of Geology, Gombe State University Nigeria,**

**[2]Exploration Geophysics, Curtin University,**

**Australia.**



## Abstract

Electrical properties of rocks are important parameters for well-log and reservoir interpretation. Laboratory measurements of such properties are time-consuming, difficult, and are impossible in some cases. Being able to compute them from 3D images of small samples will allow generating massive data in a short time, opening new avenues in applied and fundamental science. To become a reliable method, the accuracy of this technology needs to be tested. In this study, we developed a comprehensive and robust workflow with clean sand from two beaches. Electrical conductivities at 1 kHz were first carefully measured in the laboratory. A range of porosities spanning from a minimum of 0.26 to 0.33 to a maximum of 0.39 to 0.44, depending on the samples. Such range was achieved by compacting the samples in a way that reproduces natural packing of sand. Characteristic electrical formation factor versus porosity relationships were then obtain for each sand type. 3D micro-computed tomography images of each sand sample from the experimental sand pack were acquired at different resolutions. Image processing was done using global thresholding method and up to 96 sub-samples of sizes from $(200)^3$ to $(700)^3$ voxels. After segmentation, the images were used to compute the effective electrical conductivity of the sub-cubes using a Finite Element electrostatic modelling. For the samples, a good agreement between laboratory measurements and computation from digital cores was found, if the sub-cube size REV is reached that is between $(1300\mu m)^3$ and $(1820\mu m)^3$, which, with an average grain size of $160\mu m$, is between 8 and 11 grains. Computed digital rock images of the clean sands have opened a way forward in getting the formation factor within a shortest possible time; laboratory calculations take five (5) to thirty-five (35) days as in the case of clean and shaly sands respectively, whereas, the digital tomography takes just three (3) to five (5) hours.

## 1 Introduction

Electrical formation factor (FF) refers to the ratio of the electrical resistivity of a saturated medium (sediment or rock) to that of the saturating fluid (Guéguen and Palciauskas 1994). This is an important parameter in exploration geophysics as, contrary to electrical resistivity of reservoirs that is dependent on the resistivity of the saturating fluid (and hence a same type of reservoir can exhibit high or low resistivities (Mitsuhata, Uchida et al. 2006, Constable and Srnka 2007, Jinguuji, Toprak et al. 2007), formation factor is an intrinsic property of the rock, independent of fluid salinity. Measurement of formation factor in the laboratory is often



difficult and time-consuming, if not impossible in some cases. Minerals forming the rock or
sediment sample must reach thermodynamical and electrical equilibrium with the saturating
fluid, which typically takes 4 to 6 days in a high permeability high porosity clean sandstone
but may require at least 4 to 6 weeks for a tight gas sand or a low porosity rock or sediment
with a high clay content. Furthermore, results are affected by current leakage problems
(especially at high frequencies) or electrode polarization (emphasised at low frequencies).

Hence, computation of electrical properties from microstructural models has been
investigated by several teams in the past 50 years. Various methods have been proposed,
from statistical models used to reconstruct 3D porous materials e.g. (Miller 1969, Joshi 1974,
Milton 1982, Torquato 1987, Adler, Jacquin et al. 1990, Adler, Jacquin et al. 1992, Yeong
and Torquato 1998) to direct measurement of a 3D structure from synchrotron and X-ray
computed microtomography (XRCM) e.g. (Dunsmuir, Ferguson et al. 1991, Spanne, Thovert
et al. 1994, Arns, Knackstedt et al. 2001, Øren and Bakke 2002, Øren, Bakke et al. 2007,
Nakashima and Nakano 2011)  or laser confocal microscopy (Fredrich, Menendez et al.
1995). In most of these studies using XRCM images, the numerical prediction of electrical
conduction conductivity underestimates the experimental results by 30 to 100% (which leads
to an overestimation of the formation factor) (Schwartz, Auzerais et al. 1994, Spanne,
Thovert et al. 1994, Auzerais, Dunsmuir et al. 1996). Several explanations have been put
forward to justify such discrepancy: percolation differences between model and real material,
mainly to a smaller volume sampling in the model (Adler, Jacquin et al. 1992, Bentz and
Martys 1994); the addition of a third phase to the traditional two-phase model (rock matrix
being one phase and the saturating fluid being a second phase) that counts for the bound fluid
at the grain fluid interface (Zhan and Toksoz 2007); discretization errors and statistical
fluctuations (Arns, Knackstedt et al. 2001).

The underlying question behind the computation of electrical properties of digital porous
media samples (or any other rock or transport properties) is whether the obtained numerical
values are accurate One aspect of this question relates to the technology itself, namely 3D
imaging, image processing and segmentation, the suitability and stability of the numerical
code. These three key elements of the technology have been investigated by various teams
and the most comprehensive and exhaustive study performed on the various steps of the
digital rock physics workflow is the benchmark comparison from (Andrä, Combaret et al.
2013, Andrä, Combaret et al. 2013). As they are using various rock types, processing and
computing methods, the comparison is complex: they concluded that the computed effective





rock properties are affected by segmentation processes, choice of digital sub-volume, and choice of numerical code and boundary conditions. Nonetheless, the different values obtained for the formation factor deviated at most by 23% from the midrange value (Andrä, Combaret

et al. 2013). For the sphere pack sample, all computed formation factors ranged from 4.3 to 4.8.

The second aspect of this question relate to the comparison of the computed values with laboratory scale experimental data to validate the correctness of the digital rock physics workflow. However, because both experiments are done at different scale (cm scale for the

laboratory and mm scale for the digital computation), and because rocks are heterogeneous at all scales, the laboratory measured and digitally computed do not have to match. Instead, trends between two properties (e.g. formation factor and porosity) computationally derived and produced in the laboratory should be in good agreement (Dvorkin, Derzhi et al. 2011, Andrä, Combaret et al. 2013).

In the work described in this paper, we propose a robust workflow to digitally compute electrical properties of clean (i.e. that does not contain any clay or other conductive minerals) unconsolidated porous media. We first carefully measure in the laboratory the formation factor of two beach sand samples of similar mineralogy (quartz and carbonate) but of different grain size, over a wide range of porosities obtained by compacting the sand sample:

hence formation factor versus porosity trends reproducing a packing as close as possible as the one found *in*-situ were obtained. We then compute the formation factor from X-ray microtomography images using the free software finite element electrostatic code from NIST (Garboczi+++), using multiple sub-samples of various sizes. To our knowledge, this is the first time that such a work is done on clean sand.


## 2 Materials and laboratory methods

### 2.1 Sample collection and preparation

The samples investigated in this paper are sand samples collected from the coastal margin of

the Perth basin, Western Australia. The Perth Basin is an elongate, North-South trending trough underlying approximately 100,000 square kilometres of the Western Australian margin. Sediments were shed from the adjacent Yilgarn block. The Yarragadee and





Leederville sandstone formations are intercalated with the Tamale limestone that forms the Carbonates at the Upper Cretaceous. One sample was collected from Scarborough beach (31°53'41.97 S, 115°45'17.74 E) and one from Cottesloe beach (31°59'40.62 S, 115°45'03.70 E). All the samples are composed of quartz and carbonate. Grain size was determined by micro CT-image analysis and is between 16μm - 794μm (median 140.4μm) and 17μm - 606μm (median 124.0μm) for Scarborough and Cottesloe beaches, respectively. Sand samples were thoroughly washed clean with tap water to remove any plants and grass debris. Loose moist sand was then packed into the different cells used to perform the electrical resistivity measurements, then forming an initially high-porosity loose random pack; decreasing porosity in subsequent experiments was achieved by shaking the cell and using tied sticks to compact the sand: this was done in a way to achieve a packing as close as possible as the one found *in-situ*. A range of 6 different porosities were obtained for the Scarborough beach sand samples, with an initial porosity of 0.40 (loosely packed) down to 0.27 when highly packed, while 5 and 4 different porosities were obtained for the Cottesloe beach sand, depending on the geometry of the cell, with the loosely packed sample having a porosity of 0.39 and the highly packed sample having a porosity of 0.30.

Porosity was determined from the weights and densities of the sand grains and the known volumes of cells used in the experiment, as:

$$\phi = \frac{\left(V_t - m/\rho\right)}{V_t} \tag{1}$$

where $\phi$ is porosity, $V_t$ is the total volume of the cell, $m$ is the average mass of the dry sand before and after the experiment and $\rho$ is the density of the sand grains. Grain density was measured by pycnometry and found to be equal to 2.71 g/cm$^2$.

**2.2 Laboratory set-up and measurements**

**2.2.1 Experimental set-up**

Two different types of cells are used in the experimental set-up that was utilised to monitor the electrical resistivities of the sand samples as a function of salinity of the saturating pore water. These two experimental set-ups are schemed in Figures 1 and 2. For the cell called "flow cell", sample's electrical resistances are measured while saline solutions of increasing salinities are continuously flooded through the sand samples. Before proceeding with the next



saline solution, the reading of the sample's electrical resistance is let stabilize for a few hours. For the cell called "static cell", the sand samples are successively saturated with saline solutions of increasing salinities, let equilibrate with no fluid flow until stability of the sample

electrical resistance reading is achieved, and then drained before saturating the sand sample with the next saline solution. The flow cell is of cylindrical shape, 27 cm in length and 5 cm in radius (total volume of 2,120.6 cm$^3$) while the static cell is of rectangle shape, 29.8 cm in length, 8.7 cm in width and 6.2 cm of height (total volume of 1,607.41 cm$^3$).

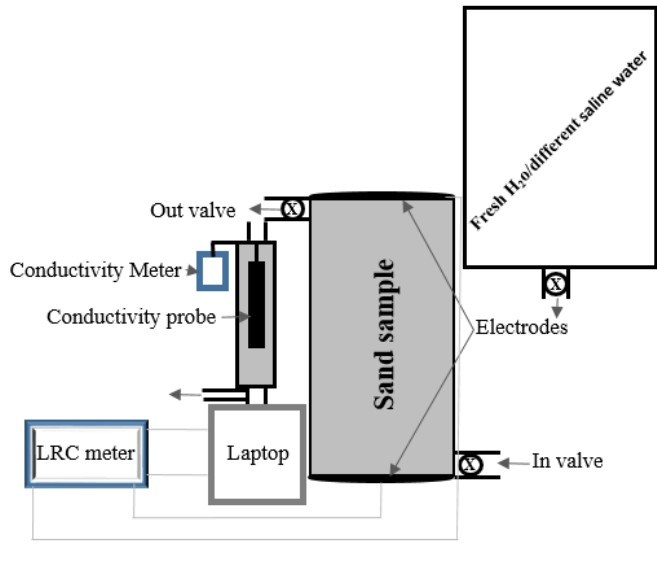

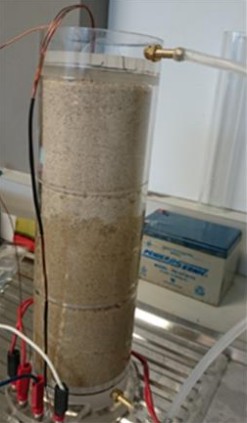

Fig 1: Schematic drawing of the experimental set up (flow cell)




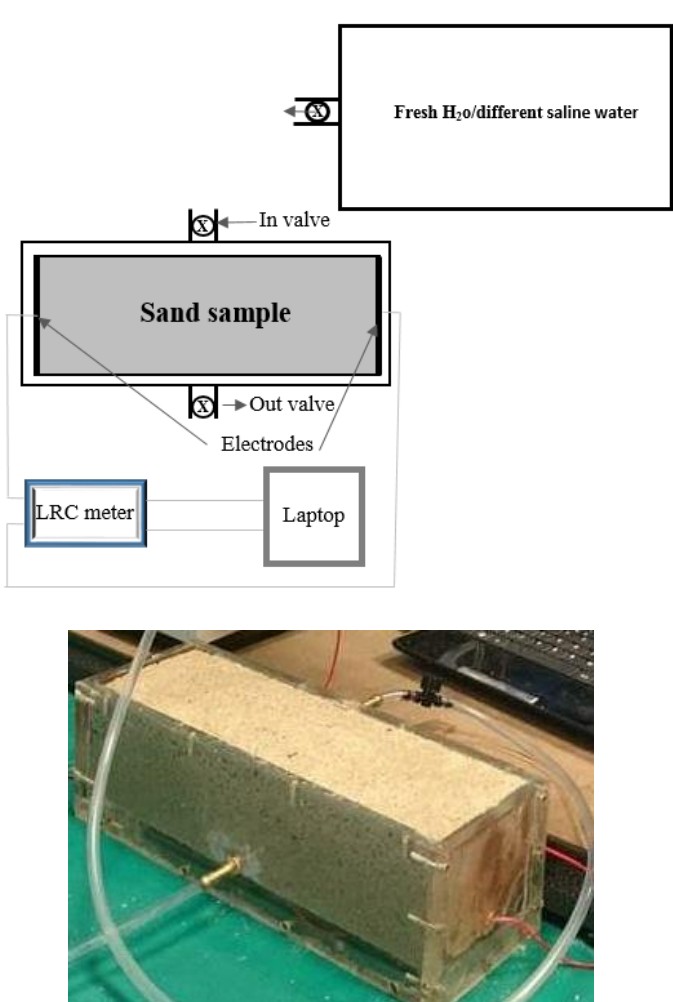

Figure 2: Schematic drawing of the experimental set up (Static cell)

Both cells are made up of Perspex (Acrylic) and have an outlet and an inlet connected by
        tubing to a tank that serves as reservoir for the various solutions injected into the sand
        samples. The solutions flow through the sand samples via gravity (falling-head method) and,
        for the flow cell, two valves, at the inlet and outlet, are used to achieve a flow rate ranging
        from 0.52 to 2.75 ml/s. This flow rate is continuously recorded.


        Injected solutions are fresh and saline solutions made with tap water and table salt in various
        amounts: 5 different salinities of 0g/L, 5g/L, 15g/L, 25g/L and 35g/L were made; both were



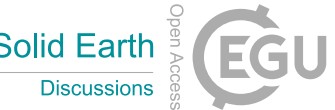

measured on an electric balance (*Napco JA-5000*) and the solution was stirred until complete dissolution of the salt into water.

Both cells are equipped with two electrodes made of zinc wire gauze with surface areas of 78.55 cm$^2$ and 53.94 cm$^2$ for the dynamic and static cells, respectively. The electrodes are glued at the bottom and at the lid cover of the cylindrical dynamic cell while they are fixed on both sides of the rectangular static cell; the two electrodes of each cell are connected to a LCR meter (*Stanford research System SR720*), connected itself to a laptop to monitor the

electrical resistance of the sand sample; recording time interval for the dynamic cell laboratory measurements is taken at 1 minute interval while the recording time interval for the static cell laboratory measurement is 10 minutes. A drive voltage of 1 Vrms is applied and a frequency of 1 kHz is chosen to minimize the phase angle between voltage and current (i.e. electrode polarization): with these conditions, the monitored $Q$ factor did not exceed

0.095 indicating the system is nearly purely resistive. For the dynamic cell laboratory measurements, the conductivity of the injected solutions coming out of the cell is monitored by an encased conductivity meter (Hanna edge) attached to the cell at intervals of 1 minute, to make it synchronous with the sand sample resistance measurements. The fluid electrical conductivity for the static cell set-up is measured with the same probe using the saturating

solution drained from the sand sample once the resistance has become stable.

### 2.2.2 Computation of electrical formation factor

Because the sand samples do not contain any clay and because the injected solutions have a conductivity ($10^{-2}$ to $5.0\ 10^{+1}$ S/m) much larger than that of quartz or carbonate surface conductivity ($5.4\ 10^{-3}$ S/m (Miller, Bradford et al. 1988), and $1.4\ 10^{-3}$ S/m (Vialle, 2008)

respectively), surface and matrix electrical conductivities can be neglected (e.g. Johnson and Sen, 1988; Garrouch and Sharma, 1994); the electrical formation factor $F$ is then given by

$$F = R_s R_w \tag{1}$$

with

$$R_s = r_s \frac{A}{L} \tag{2}$$

$$R_w = \frac{1}{C_w} \tag{3}$$



where $R_s$ is the resistivity of the sand sample saturated with water, $R_w$ is the resistivity of the water, $r_s$ the measured resistance of the sand sample saturated with water, $A$ the surface area of the electrode, $L$ the length of the cell and $C_w$ the measured conductivity of water.

To obtain the formation factor, the sample's resistivity, once it has stabilized, is plotted against the saline water's resistivity, and the formation factor is given by the inverse of the slope.

## 3 Digital rock samples and computation of electric properties

### 3.1 Image acquisition

Two samples were prepared for imaging with X-Ray Micro-Computed Tomography (XRMCT), one from Scarborough beach and one from Cottesloe beach. Loose sand was put in a cylindrical Pyrex glass tube of 6 mm in diameter and 6 cm in height, and the tube was inserted in the core holder of the micro-tomograph. The samples were scanned with the 3D X-ray Microscope Versa XRM 500 (Zeiss – XRadia) using an X-ray energy of 60keV, a current of 70.66 µA and an energy of 5W. In each scan 3000 projections (radiographs) were acquires. The exposure time was 2s per radiograph. Initial cone-beam 3D image reconstruction was performed using the software XM Reconstruction (XRadia). A secondary reference was required to remove geometrical artefacts during reconstruction. After 3D reconstruction, 3D volume was sliced onto 2D images for further processing. A total number of 1021 2D images for Scarborough beach sample and 991 2D images for Cottesloe beach were available for analysis. Total scanning time was 2hrs 55minutes and 2hrs 42minutes for Scarborough and Cottesloe samples respectively. A nominal voxel sizes of $(2.5761\mu m)^3$ and $(2.5516\mu m)^3$ was achieved with a source-to-sample and detector-to-sample distances of 11mm and 22mm, for both Scarborough and Cottesloe beach samples respectively.

### 3.2 Image processing

### 3.2.1 Image filtering

We used the software package Avizofire 9 (FEI Visualization Sciences Group) for image enhancement and segmentation. Grey-scale images of the 2D slices were processed using a non-local filter in the intensity range of 255 – 5344 for Scarborough beach and 255 - 5467 for Cottesloe beach, with the aim of removing concentric shadows in the images and properly





enhancing interfaces between the pores and grains as well as removing noise. Figures 3(a)-3(d) shows raw and filtered images for both Scarborough and Cottesloe beach: we can easily notice that the quality of the image has increased. In these images, the white grains are carbonate, grey grains are quartz, while black within the cycle corresponds to void space (pores).

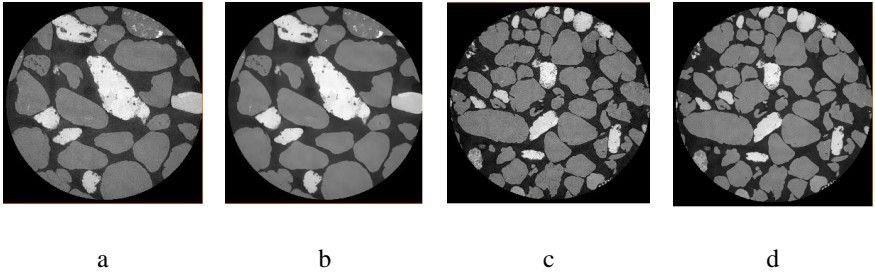

a                      b                      c                      d

Figure 3: a) Raw and b) filtered images of Scarborough beach sand sample; c) Raw and d) filtered images of Cottesloe beach sand sample.

**3.2.2 Image segmentation**

The filtered images were segmented using two types of thresholding algorithms: the first one resulted in a 2-phase segmentation that was further used for computing samples electrical conductivities; the second one is a watershed algorithm that resulted in a 2- or 3-phase segmentation used for grain analysis. Note that filtering and segmentation workflows were
applied to the full 3D dataset. Figure 4 shows the histogram for both samples.

*2-phase segmentation by global thresholding*

 Because both quartz and carbonate have very low conductivity compared to that of water, they can be both considered as non-conductive for computation purposes of electrical conductivity of the water-saturated sand sample. Hence quartz and carbonate can be put in a
single phase, and pores will constitute a second phase, that will be later on filled with a conductive fluid for the computation of sample electrical properties. We use here a global threshold segmentation algorithm to separate pores from grains: the set intensity value separating pores from grains (both quartz and carbonate grains having higher intensity values than that of pores) is kept the same for all 2D slices.



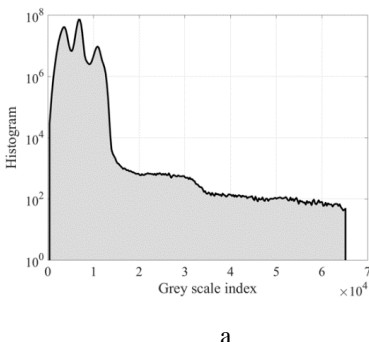
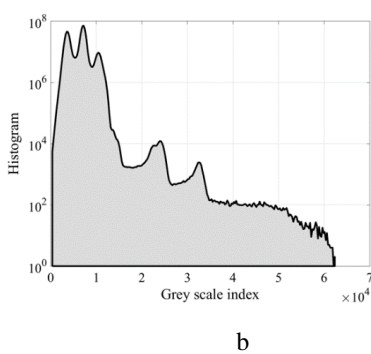

a                                        b

Figure 4: Histogram of (a) Scarborough and (b) Cottesloe beaches.


Poor segmentation can affect accurate calculation of porosity. To check the quality of the
segmentation, we compare the porosity estimated in the laboratory with the one estimated
from micro CT-scan images. We made a random loose pack of sand (cm$^3$) in the laboratory to
obtain the highest porosities of 0.361 and 0.349 from Scarborough and Cottesloe beaches
respectively while the smaller scanned sample of the sand (mm$^3$) was also randomly packed
in the small tube from which porosities of 0.369 and 0.359 were obtained from the images of
Scarborough and Cottesloe beaches respectively.

***Watershed segmentation***

We used a marker based watershed segmentation algorithm from Avizo Fire 9. We defined
either 2 or 3 marker ranges of grey scale intensity for either, pore and grains, or for pore,
carbonate grains and quartz grains, respectively. We then performed a watershed flooding for
each of these 2 or 3 phases. The 2-phase watershed segmentation allows computation of pore
volume and grain size distribution, whereas the 3-phase segmentation (figure 5) gives volume
fraction of the different minerals.





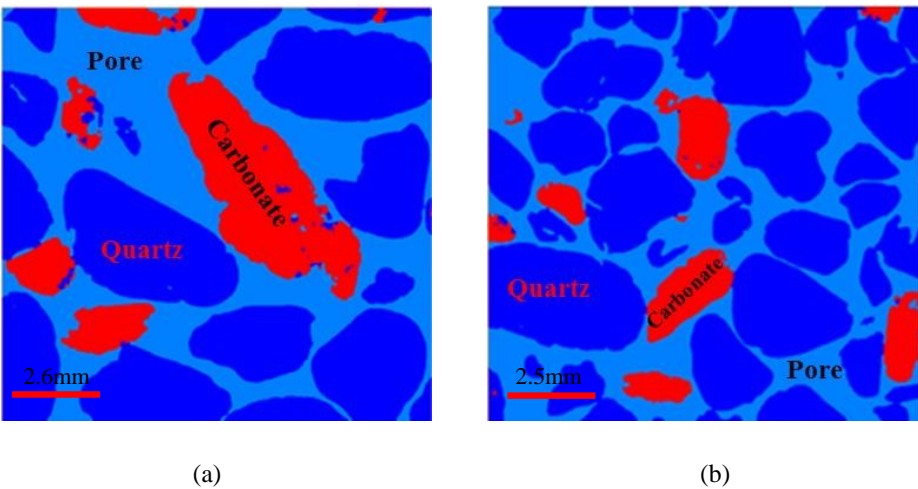

(a)        (b)

Figure 5: 3-phases watershed segmentation of the sand samples a) Scarborough; b) Cottesloe


### 3.2.3 Image cropping

The 3D filtered and segmented volumes for each of the two sand samples were subdivided into overlapping sub-cubes (96 in total) of 4 different sizes: 3 sub-cubes of a size of $(700)^3$, 8 of a size of $(500)^3$, 13 of a size of $(350)^3$, and 20 of a size of $(200)^3$ for Scarborough beach sample, and 5 sub-cubes of a size of $(700)^3$, 10 of a size of $(500)^3$, 13 of a size of $(350)^3$, and 24 of a size of $(200)^3$ for Cottesloe beach sample. Porosity was estimated using Avizo software for each of these 96 sub-cubes.

The 2D cropped images were then exported in binary format for computation of electrical properties.

### 3.3 Computational studies of electrical fields of micro-CT images

To estimate conductivity from micro-CT images, we assume that pores are electrically conductive, and that the solid phases are not conductive. This assumption is based upon the concept that mainly the ions in fluid-filling pores can be drifted under the effect of external electric fields. To estimate the conductivity from images, first, we have to calculate an average current density.




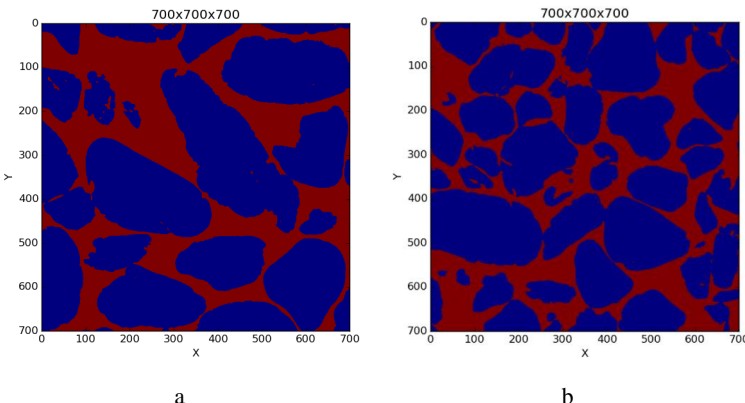

Figure 6: 700 binary images (a) Scarborough and (b) Cottesloe beaches.

If we assume that the conservation of charge is valid in the pore structure, then no net charges are created or annihilated in the pore volume and pore surfaces; the current density vector obeys the following equation:

$\nabla . J = 0.$    (4)

On the other hand, Ohm's law at the microscopic level assumes that the current density is proportional to the electric potential field:

$J = \sigma_w \nabla V$    (5)

where $J$ is the electrical current density, $\sigma_w$ is the electrical conductivity of the fluid that fills the pore space, $V$ is the electrical potential field (voltage). By substituting Eqn. (5) into Eqn. (4), we have the Laplace equation as:

$\nabla \cdot \left( s_w \nabla V \right) = 0$    (6)

Eqn. (6) can be solved numerically for pore structures by applying an external electric field

$\vec{E}_{ext}$ on the boundaries. One of most reliable numerical methods to estimate the average current density from 3D images is the finite element method. We use the freely available code written by (Garboczi, E. J. 1998). This code, by minimizing the electrical energy stored in the porous volume under study, estimates the local potential field (V) at each coordinate system (pore and solid phases). For a giving microstructure, because of the applied fields or

other boundary conditions, the final voltage distribution is determined by minimization of the





total energy stored in the system. Figures 7a and 7b show the potential field variations in Scarborough and Cottesloe beach samples, respectively. This can help us evaluate the effective current density ($\vec{J}_{av}$) by using equation (7) and by taking the volume average of the local current density vectors ($\vec{J}$). On the other hand, the volume average of current density is

defined as:

$$\vec{J}_{av} = \langle \vec{J} \rangle = \sigma_{eff} \vec{E}_{ext} \tag{7}$$

where $S_{eff}$ is the effective conductivity of the porous medium. Effective conductivity is a 2$^{nd}$ rank tensor. In Equation (7), the current density ($\vec{J}_{av}$) and the external electrical field ($\vec{E}_{ext}$) are vectors. If we assume that the external electrical field is unidirectional (let assume in the

x-direction, $\vec{E}_{ext} = E \cdot \vec{u}_x$) then the current density can have components on any other directions and can be thus written in the general form as:

$$\vec{J}_{av} = J_x \cdot \vec{u}_x + J_y \cdot \vec{u}_y + J_z \cdot \vec{u}_z \tag{8}$$

Then, from Eqn. (7), the current density can be rewritten as:

$$\vec{J}_{av} = \sigma_{xx} E \cdot \vec{u}_x + \sigma_{yx} E \cdot \vec{u}_y + \sigma_{zx} E \cdot \vec{u}_z \tag{9}$$

In homogenous media, we expect the current density to be negligible in the direction perpendicular to the external electrical field. This implies that for homogenous media, the effective conductivity tensor is a diagonal matrix. On the other hand, for heterogeneous media, the current density in the direction perpendicular to the external electrical field is not zero, or is not small compared to the diagonal values. Hence, in general, the current density is

second rank tensor of the form:

$$S = \begin{pmatrix} S_{xx} & S_{xy} & S_{xz} \\ S_{yx} & S_{yy} & S_{yz} \\ S_{zx} & S_{zy} & S_{zz} \end{pmatrix} . \tag{10}$$



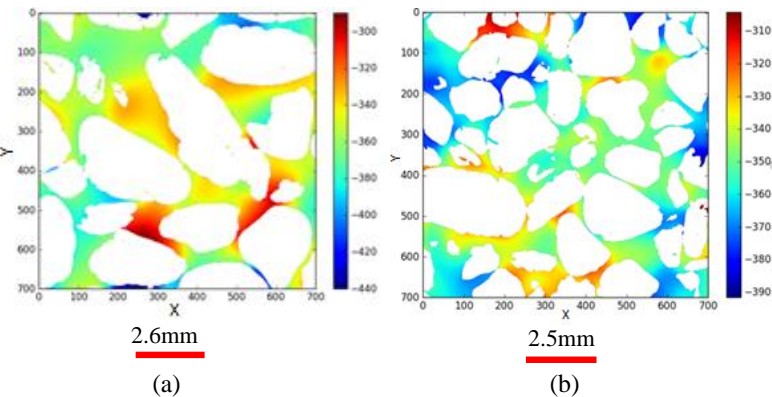

Fig 7: Electrical potential fields image output from the $(700)^3$ digital sub-cubes of (a) Scarborough (b) Cottesloe beaches.

The $(700)^3$ voxel cube from Scarborough sample was analysed by applying a current successively in x, y and z-directions to find out whether the sample shows some anisotropy.

The output of conductivity along x, y and z-directions shows almost the same values of formation factor (5.30, 4.96 and 5.08 respectively). The difference in the values of formation factor between the x-direction and y-direction is 0.033% while that between the x-direction and z-direction is 0.021%; hence, the sample is isotropic in nature, at the scale of investigation. For sure by decreasing the size of the images, the pore structures became more

anisotropic, but on the other hand the volumes investigated are not representative any more. Note that we discuss in more details the concept of representative elementary volume (REV) in section 4.2. Therefore, by having isotropic conductivity of the representative elementary volume, here and after, we just assume the isotropic pore structures and consider the conductivity as a scalar number for all images.

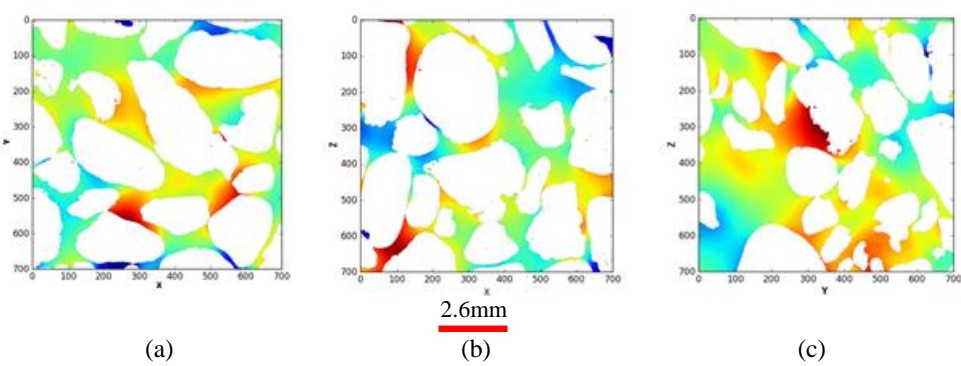



Figure 8: Electrical potential fields images (a) along x direction, (b) along y direction and (c) along z-axes.


From the effective conductivity calculated for micro-XRCT images, the electrical formation factor can be estimated as:

$$F = \frac{S_w}{S_{eff}},$$  11

where $S_w$ is the electrical conductivity of pore fluids, taken equal to 1 in the computation.

Electrical formation factor is calculated for each of the different sub-cubes obtained from the micro-CT images of Scarborough and Cottesloe beach samples.

## 4. Results

### 4.1 Laboratory

Figure 9 displays the values of formation factor trend against porosity for Scarborough and

Cottesloe beaches respectively, computed as described in section 2.2.2 and for each porosity value obtained by compacting the initial sand pack. The results for both 'static' and 'flow' cells are reported in Tables 1 and 2 for both samples, and for all data points. The values of formation factors obtained using the 'flow' cell are higher than that obtained using the 'static' cell for both Scarborough (8.2) and Cottesloe (8.5) beach samples, whereas for Scarborough

beach, formation factors have close values at high porosities and then depart from each other at lower porosities (from lower than 0.39). In these figures, we have bounded the experimental data by two lines that represent a power-law relationship between the formation factor and porosity in the form

$$F = a \times f^{\,m} = f^{\,m}$$  12

This is Archie's law (Archie, 1942) with a tortuosity factor *a* of 1. Tortuosity factor usually ranges from 0.5 to 1.5, and though there has been quite a wide range reported in literature for sand, from the most used value of 0.62 (Humble formula, Winsauer et al., 1952) to up to 2.45 (Carothers and Porter, 1970). We take here the same tortuosity factor value of 1 for all samples. This is the value for clean granular formations (Sethi, 1979).






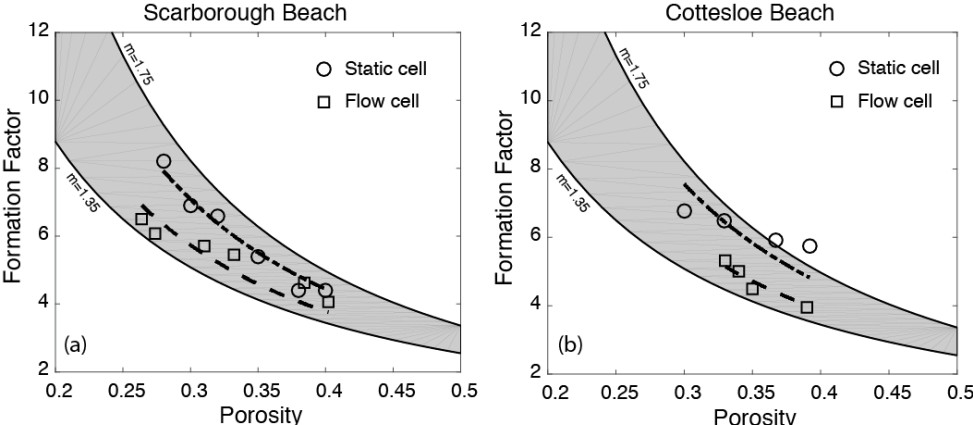

Figure 9: Laboratory measured formation factor versus porosity values for both flow and static cell for (a) Scarborough and (b) Cottesloe beach samples.

## 4.2 Micro CT-scan images

Formation factor were plotted against porosity for all the micro CT-scan image cubes for Scarborough and Cottesloe beaches (Figures 10 and 11, respectively).

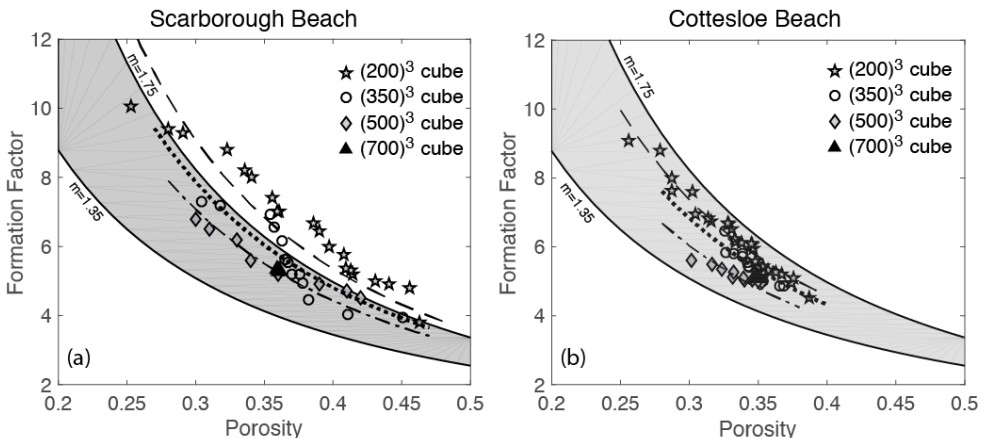

Figure 10: Formation factor against porosity for each sub-cube size of, $(200)^3$, $(350)^3$, $(500)^3$ and $(700)^3$ from both (a) Scarborough beach samples and (b) Cottesloe beach samples.

Similarly, both porosity and formation factor were plotted against the cube sizes $200^3$, $350^3$, $500^3$ and $700^3$. Scattering is shown when the cube sizes were small which begin to tapered as



the Representative Elemental Volume (REV) is approached. This REV is somewhere between $(500)^3$ and $(700)^3$, which corresponds to a sample size between $(1.3mm)^3$ and $(1.8mm)^3$.


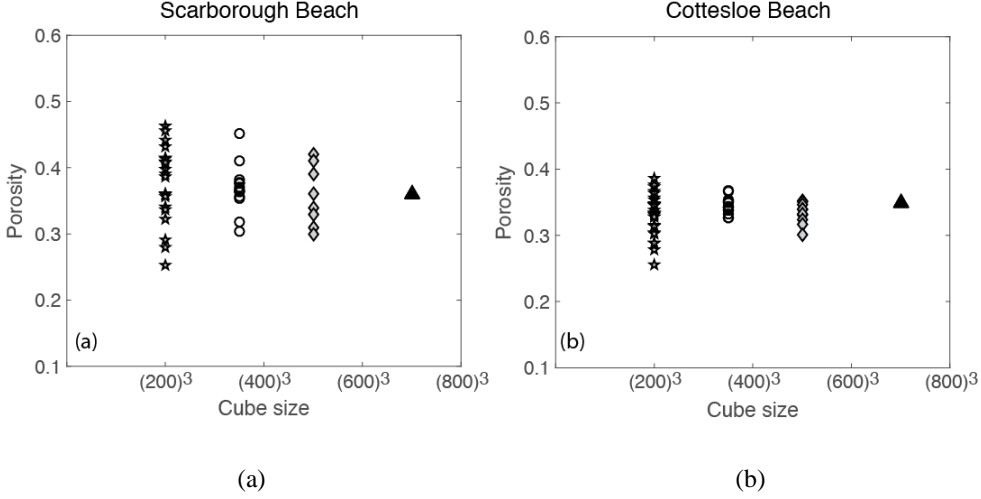

(a)                                        (b)

Figure 11: Porosity against cube sizes (a) Scarborough beach (b) Cottesloe beach.

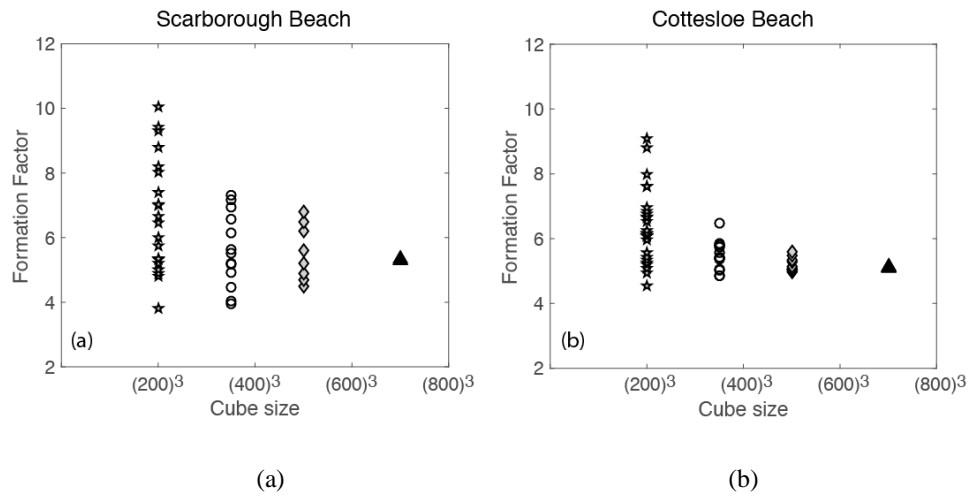

375                          (a)                                        (b)

Figure 12: Formation factor sizes (a) Scarborough beach (b) Cottesloe beach.






Table 1: Summary of Laboratory and micro-CT scan images result from Scarborough beach samples.

| Group | Sample | Measure | Values |
|---|---|---|---|
| Laboratory results | Flow cell | Porosity | 0.40 0.38 0.35 0.32 0.30 0.27 |
| | | F.F | 4.4 4.4 5.4 6.6 6.8 8.2 |
| | Static cell | Porosity | 0.40 0.38 0.33 0.31 0.27 0.26 |
| | | F.F | 4.0 4.6 5.5 5.7 6.1 6.5 |
| Micro-CT scan images | 700 cubes | Porosity | 0.36 0.36 0.36 |
| | | F.F | 5.3 5.3 5.3 |
| | 500 cubes | Porosity | 0.42 0.41 0.39 0.36 0.34 0.33 0.33 0.33 |
| | | F.F | 4.5 4.7 4.9 5.2 5.6 6.2 6.5 6.8 |
| | 350 cubes | Porosity | 0.45 0.41 0.38 0.38 0.38 0.37 0.37 0.37 0.36 0.36 0.36 0.32 0.30 |
| | | F.F | 3.96 4.03 4.46 4.93 5.19 5.21 5.53 5.63 6.15 6.57 6.93 7.18 7.30 |
| | 200 cubes | Porosity | 0.46 0.46 0.44 0.43 0.41 0.41 0.41 0.40 0.39 0.39 0.36 0.36 0.34 0.32 0.29 0.28 0.25 |
| | | F.F | 3.8 4.8 4.9 5.0 5.2 5.3 5.4 5.8 6.0 6.5 6.7 7.0 7.0 7.4 8.0 8.2 8.8 9.3 9.4 10.1 |



Table 2: Summary of Laboratory and micro-CT scan images result from Cottesloe beach samples.

| | | | | | | | | | | | | | | | |
|---|---|---|---|---|---|---|---|---|---|---|---|---|---|---|---|
| **Laboratory results** | **Flow cell** | **Porosity** | 0.39 | 0.35 | 0.34 | 0.33 | 0.26 | | | | | | | |
| | | **F. F** | 3.96 | 4.50 | 5.00 | 5.33 | 8.54 | | | | | | | |
| | **Static cell** | **Porosity** | 0.37 | 0.35 | 0.33 | 0.31 | | | | | | | | |
| | | **F. F** | 5.72 | 5.93 | 6.50 | 6.90 | | | | | | | | |
| **Micro-CT scan images** | **700 cubes** | **Porosity** | 0.35 | 0.35 | 0.35 | 0.35 | 0.35 | | | | | | | |
| | | **F. F** | 5.1 | 5.1 | 5.1 | 5.1 | 5.1 | | | | | | | |
| | **500 cubes** | **Porosity** | 0.35 | 0.35 | 0.34 | 0.34 | 0.34 | 0.33 | 0.33 | 0.33 | 0.32 | 0.30 | | |
| | | **F. F** | 4.97 | 5.01 | 5.02 | 5.04 | 5.09 | 5.13 | 5.27 | 5.34 | 5.48 | 5.59 | | |
| | **350 cubes** | **Porosity** | 0.368 | 0.366 | 0.353 | 0.352 | 0.351 | 0.349 | 0.344 | 0.343 | 0.342 | 0.338 | 0.332 | 0.327 | 0.326 |
| | | **F. F** | 4.87 | 4.87 | 5.01 | 5.07 | 5.29 | 5.36 | 5.39 | 5.42 | 5.55 | 5.73 | 5.80 | 5.84 | 6.47 |
| | **200 cubes** | **Porosity** | 0.39 | 0.37 | 0.37 | 0.36 | 0.36 | 0.35 | 0.35 | 0.35 | 0.35 | 0.34 | 0.34 | 0.33 | 0.33 |
| | | **F. F** | 4.5 | 4.9 | 5.2 | 5.3 | 5.3 | 5.4 | 5.6 | 6.0 | 6.1 | 6.1 | 6.1 | 6.1 | |
| | | **Porosity** | 0.33 | 0.33 | 0.32 | 0.31 | 0.30 | 0.30 | 0.29 | 0.29 | 0.28 | 0.26 | | | |
| | | **F. F** | 6.2 | 6.5 | 6.6 | 6.7 | 6.7 | 6.8 | 7.0 | 7.6 | 7.6 | 8.0 | 8.8 | | |






## 5. Discussion

Presented in tables 1 and 2 are the values of formation factor and porosity obtained from both laboratory experiments and micro CT scan images for Scarborough and Cottesloe beach samples.

As noticed earlier in section 4.1, the values of formation factor obtained by the static cells are higher than that obtained by the dynamic cell (for a given porosity), for both samples. This translates in a higher cementation exponent $m$. One reason for this can be the design of the cell itself and of the way to achieve a stable reading of sample conductivity, for each fluid salinity. In the rectangular (static) cell, because the higher salinity brine is introduced or

retrieved via the centre of the panels (see Figure 2) there could some brine left in the corners that will only equilibrate with the new injected brine by diffusion and hence there could be a lower conductivity of the brine in these corners compared to the conductivity of the injected brine. As result the measured sample conductivity will be lowered with respect with what it should be, giving a higher ratio sample to brine conductivities (i.e. formation factor, see Eqn.

11). Using a cylindrical cell has thus the advantage of providing a better replacement of the brine.

The formation factor trend of laboratory measurements (flow and static) are in agreement with the formation factor from the micro-CT scan, however, best agreement was observed when the cube size reaches the Representative Elemental Volume (REV) at about 500 and

350 cubes for Scarborough and Cottesloe sand beach respectively.

The porosity and formation factor of 2 samples one each from Scarborough and Cottesloe beaches (Scarborough having 5 different porosities) and (Cottesloe having 4 different porosities)  all the nine (9) porosity ranges between 0.26 – 0.40 which fall within the range for heterogeneous sands as reported by (Salem 2001). While formation also has a value range

between 03.95 – 8.20 for both sand samples and the range found is within the one found in literature (Salem and Chilingarian 1999). (Erickson and Jarrard 1999) investigated siliciclastic sediments from an in-situ core data of Amazon submarine fan; they compared the trends of formation factor against porosity for both clean sand and that of sandy clay and conclude that formation factor depends on porosity and lithology.  In addition, that in high-

porosity sands, presence of clay reduces formation conductivity by increasing the tortuosity. Researchers have given little attention to pore geometry and Archie cementation factor, but (Salem 2001) studied the glacial deposits of silts, sand and gravels from the freshwater





aquifer of Northern Germany in which he placed emphasis on both pore geometry and Archie formation factor were he obtained a porosity range of 0.25 - 0.51 with a corresponding

formation factor range of 4 - 16.

(Luquot, Hebert et al. 2016) also found a perfect agreement of both structural and geometrical parameters calculated from laboratory and X-ray microtomography of Limestone from Dordogne region of France. (Saenger and Kuhs 2016) worked on gas hydrate from mixture of sedimentary matrices of natural quartz from Moscow with small quantity of montmorillonite

clay from Turkenista with glass beads. In their research, they got a range of porosity of 0.31 – 0.55 with a formation factor ranging 4.5 – 5.1 and conclude that laboratory results do not usually converge when experiments were performed on a more matured natural sample.

Table 1 and 2 shows the formation factor of 96 cubes, 44 cubes from Scarborough with a range of formation factor of 4.0 – 8.2 (laboratory measurement), 3.80  - 10.1 (Micro CT-scan)

and 52 cubes from Cottesloe  having a range of formation factor of 3.96 – 6.74 (laboratory measurement), 4.50  - 8.80 (Micro CT-scan), all samples falls within the range found in literature, porosity also has a value range between 0.26 – 0.40 and 0.30 – 0.39 (Laboratory measurements) for Scarborough and Cottesloe respectively while a porosity of 0.25 -0.46 and 0.26 – 0.39 (Micro CT-scan images) for Scarborough and Cottesloe beaches respectively

(figure 16).  All values fall within the range for heterogeneous sands as reported by (Salem 2001) except three (2) 200 cubes and one (1) 350 cubes that are above the porosity reported and this is because these cubes are far below the Representative Elemental Volume and does not represent the sample.



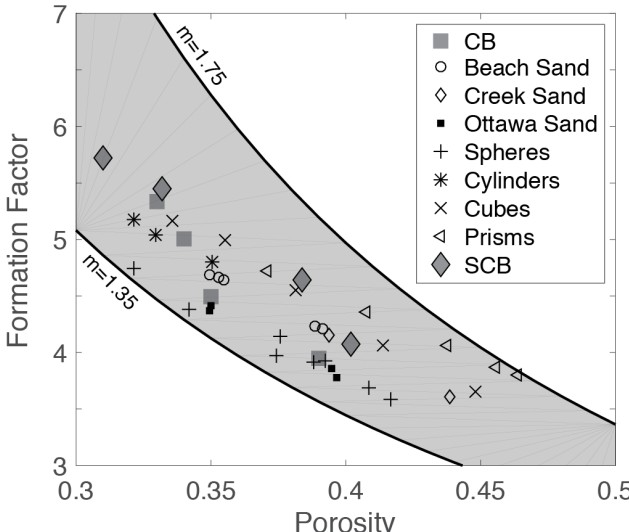

Figure13: Summary of Laboratory and micro CT-scan results with results from other
workers.

**6. Conclusions**


Electrical properties of rocks are important parameters for well log and reservoir
interpretation. Laboratory measurements of such properties are time-consuming, difficult, if
not impossible in some cases. In view of this, we have successfully combined the scientific
approach of laboratory measurements (as a bench mark) with micro-CT scan computational

images and have achieved the objectives of computing the variability of computed formation
factor as a function of porosity from laboratory measurements and micro-CT scan images
from 2 sand samples of Scarborough and Cottesloe beaches of Perth basin, for fastest method
of obtaining the formation factor from CT-scan images that takes shorter time (5-7 hours)
with calculation from laboratory measurements that takes much more longer time (30-

65days).

This approach is practical, easily repeatable in real time (though expensive) and can be an
alternative method for calculating formation factor when time is not on the side of the



experimenter, which is always the case. Results of images below $500^3$ (Scarborough) and $350^3$ (Cottesloe) beaches indicates that they are not suitable REV for pore scale networks.

In this paper, micro CT-scan images computational technique was employed to calculate properties such as porosity and formation factor on large three-dimensional digitized images of sand sample. We demonstrated that for most of the parameters studied here, the values obtained by computing micro CT-scan images agreed with the classical laboratory measurements and results from other workers. However, in further work a way of least

sample matrix disturbance during changing the porosity (by gently shaking the cell) should be achieved. The natural sediments found in our environments are heterogeneous in nature; therefore, calculating formation factor from micro CT-scan images of sandy clay mixture is recommended (work is in progress).

*Acknowledgments: We acknowledge the help of Dominic Howman, and Vassili Mikhaltsevitch for*
*their help in cell design and laboratory experiments and Andrew Squelch for help with image*
*processing.*

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
