# Peer review of "Electrical Formation Factor of Clean Sand from Laboratory Measurements and Digital Rock Physics."

_Solid Earth, 2018_

## Referee Comment (RC1) · Anonymous Referee #1 · 28 Jan 2019

This paper calculated the formation factor of two clean sand samples with multiple sub samples with various sizes and compared with the lab results. That is a meaningful job. There is just one question, for the small size sub-cubes that maybe anisotropic, how you choose the effective conductivity, just one direction or an average value?

---

## Referee Comment (RC2) · Anonymous Referee #2 · 24 Feb 2019

Electrical properties of rocks were computed from 3D images of samples and the accuracy was validated by careful laboratory measurement in this paper. The methodology is a logical and helpful way to replace time-consuming laboratory measurements. The suggestions for future work are as follows: 1) the filtering functions can be improved to avoid holes inside sand profiles ;2)the global threshold for two phase segmentation can be improved to obtain sharper outlines of sands and avoid connectivity between two sand profiles.

---

## Referee Comment (RC3) · Anonymous Referee #3 · 13 Mar 2019

In this study, the authors develop a comprehensive and robust workflow with clean sand from two beaches. Electrical conductivities are measured in laboratory. Characteristic electrical formation factor versus porosity relationships are obtain for each sand type. In other hand, 3-D micro-computed tomography images of each sand sample from the experimental sand pack are acquired at different resolutions. The images are used to compute the effective electrical conductivity of the sub-cubes using a Finite Element electrostatic modelling. For the samples, a good agreement between laboratory measurements and computation from digital cores is found with the sub-cube size REV is reached some conditions. Computed digital rock images of the clean sands opens a way forward in getting the formation factor within a shortest possible time. This topic

is welcomed in fields of reservoirs and geophysics. However, this paper is more general and should be improved in many places. 1. Besides experiments, there also are some theoretical and modeling works on electrical properties of rocks/porous media, these should be well summarized and reviewed. 2. Writing should be improved and concised. Many basic descriptions are not necessary. 3. Please discuss the limitation of your work/method, such as for tight or low permeability rocks.

---

## Referee Comment (RC4) · Anonymous Referee #4 · 3 Apr 2019

This paper reports on a study aimed at comparing the values of the electrical formation factor of natural sands, determined by classical measurements on the one hand, and determined by numerical computations on micro CT images on the other hand. This is an interesting approach: indeed, image-based computation requires less time that experiments. The paper is well-organized and well-written. I think that this study can deserve publication, after some moderate revisions.

Line 123: why do you choose to study natural sands made of quartz and carbonates? Why not pure quartz sands or pure carbonate sands first?

Line 126: what is the carbonate/quartz content (in %) of the two sands? Have the

quartz and carbonate grains the same grain size distributions?

Line 130: how are you sure that after compaction the sandpack is homogeneous?

Line 158: you should add some words about the "non-conventional" rectangular cell. Why did you use such a geometry? What was the objective of using this configuration?

Figures 1 and 2: add the scale

Line 195, equation 3: use sigma_w instead of C. C is generally used to denote the concentration, not the electrical conductivity.

Line 201: maybe show an example of sigma_rock vs sigma_water with the fitting straight line.

Table 1: maybe provide the adjustment coefficient to provide an estimation of the quality of the value of FF

Figure 7: add th unit for the electrical field.

Figure 8: if I am right, this figure is not referenced in the text. Colorbar and unit are missing.

Figure 9: could you add the error bars, for both porosity and formation factor? Also add the value of the cementation exponent for the dashed lines corresponding to the fit of the experimental data.

Line 380: to validate your approach, a figure is missing, showing the comparison of the measured and compared value. I suggest you to plot measured FF/porosity and computed FF/porosity, as well as the 1:1 line.

Discussion: again, a comment on the interest of the unconventional cell is required. A comment about the deviating trend of the measured data for the Cottesloe sand with unconventional cell is missing.

Figure 13: informations are missing in the caption. Which data are from experimentally

measured values, from image-computation? The dots corresponding to this study are missing (for comparison). Moreover, the references of the data should be provide (for instance, "from Smith et al.").

---

## Referee Comment (RC5) · Anonymous Referee #5 · 13 Apr 2019

The manuscript is about measuring and numerically computing formation factor of sand-pack samples. The problem is well known and the author did not improved it theoretically, but conducting such neat experiments and computations are interesting. The MS is well written with fluent English and proper structure, which a reader can easily follow it from A to Z. In overall, I think it deserves to be published after some corrections. My main concern is: Discussion is mostly a report than a really discussion. There are at least two questionable points, which should be considered in this part: i. difference between FF of rectangular and cylindrical cells, and ii. different trends between FF and porosity in Figure 9. For example, there are obviously another trends than FF=Phiˆ(-m) in some cases (e.g., Scarborough Beach, flow or Cottesloe

[Figure]

Beach, static), but authors insisted on Archie trend. These differences go back to the porous structure of each sample and since the authors have access to it (I mean via segmented images), they should do more extensive work in this part. They can at least calculate pore network model or tortuosity of flow paths of each sample and compare their properties to find reasonable relationships.

---

## Author Comment (AC1) · 21 May 2019

Referee Comment here is just one question, for the small size sub-cubes that maybe anisotropic, how you choose the effective conductivity, just one direction or an average value?

Authors Reply That is an excellent point. Yes, we examined the full tensor of conductivity for the 700ˆ3 samples, and it has been found that the tensor is isotropic within the small relative error. For the small samples, which are smaller than RVE, we took an average of conductivities of different directions, which mathematically is equal to one-third of the trace of conductivity tensor.

[Figure]

Changes in the text We replace the sentence L. 329 to 330 by "In the following, we took an average of the conductivities in the three different directions, which mathematically is equal to one-third of the trace of conductivity tensor; for simplicity, we then consider the conductivity as a scalar number for all images"

———————————————————

---

## Author Comment (AC2) · 21 May 2019

Referee Comment The suggestions for future work are as follows: 1) the filtering functions can be improved to avoid holes inside sand profiles; 2) the global threshold for two phase segmentation can be improved to obtain sharper outlines of sands and avoid connectivity between two sand profiles.

Authors Reply Thank you for your comment. We are indeed aware that the filtering and segmentation phases can be done in a different way and eventually improved. However we still want to keep the workflow as 'light' as possible to avoid excessive computational times while obtaining results as accurate as possible. As regard the 2-

phase segmentation method, it gives some porosities of 0.36 and 0.37 for the two types of beach sand samples, which is totally reasonable for a 'dense random pack' (∼0.36 e.g. Mavko et al., 2009) and in agreement with the experimentally measured porosities on the lab-scale samples (0.35 and 0.36), with the uncertainties of both methods. We thus considered that this segmentation method was enough for our study. As regard the filtering, our main concern was to remove the ring artefacts (and effect of the image acquisition process) and the non-local filter is indeed a good one for that purpose. Furthermore it does not introduce edge smoothing contrary to many other filters and thus does not require the use of an additional mask (see for example the review paper of Schluter et al., 2014)

Changes in the text To reflect the previous we have added a few justifications about the filter used L227: "Non-local means filter has been shown to effectively remove ring artefacts without introducing edge smoothing contrary to many other filters and thus does not require the use of an additional mask (see for example the review paper of Schluter et al., 2014)" We added the above reference too.

---

## Author Comment (AC3) · 21 May 2019

Referee Comment #1 besides experiments, there also are some theoretical and modeling works on electrical properties of rocks/porous media, these should be well summarized and reviewed.

Authors Reply Yes indeed some modelling work has been done too: we have summarized them from L.64 to L.81. We have though just focused on the computation of electrical properties from microstructural models, as this is the main object of this paper (and not on the actual theory related to electrical properties of porous medium). Besides, the paper is already quite substantial and we did not want to add too much

information that would unnecessarily lengthen it.

Referee Comment #2 Writing should be improved and concised. Many basic descriptions are not necessary.

Authors Reply We think it is important to describe with enough details the methods, techniques and computation steps, as it allows readers to see how the various results have been obtained. It is also necessary to have these information for comparing the results obtained by the various techniques or experimental devices, as some differences between them could be inherent to the methods used. Hence this is why we have preferred to have a shorter and more focused introduction and then allow for a more detailed methodology part.

Referee Comment #3 Please discuss the limitation of your work/method, such as for tight or low permeability rocks

Authors Reply. Indeed our work has been developed for non-conductive, unconsolidated materials and would require further development for other types of rocks or materials. However, tight or low permeability rocks are only a subset (and in that case the most challenging part may be the imaging itself as a classical micro-CT may not resolve the small pore and pore throats sizes). We thus added the following in the conclusion part.

Changes in the text L470 "This work was focused on establishing a robust methodology and workflow and we thus started with one of the most simple materials, though still highly relevant for many applications in oil & gas or water management environments. For more complex geological materials, such as low-permeability rocks, multi-mineralitic rocks, materials with conductive minerals, etc., further developments are obviously needed. However these developments are mostly related to the employed techniques (e.g. a higher-resolution imaging technique would be need for low-permeability rocks, a more complex laboratory set-up and techniques for measurements of rocks with conductive minerals or minerals with a non-negligible surface conductivity, etc.)

rather to the overall workflow established here (comparison between laboratory and computed data through trends between properties) that remain valid."

---

## Author Comment (AC4) · 21 May 2019

Referee Comment #1 Line 123: why do you choose to study natural sands made of quartz and carbonates? Why not pure quartz sands or pure carbonate sands first?

Authors Reply These sands are the sands found in the Perth Basin and so this was driven by a practical aspect. Furthermore, within our department, we work on projects that involve electrical resistivity surveys of the coastal area, and thus it was appropriate to perform some laboratory work in relation to these projects.

Authors changes none

Referee Comment #2 Line 126: what is the carbonate/quartz content (in %) of the two sands? Have the quartz and carbonate grains the same grain size distributions?

Authors Reply Thank you for this comment as this is indeed a needed information. We completed the sentences L. 126 as follow.

Authors changes Line 126: "All the samples are composed of quartz and carbonate, in a proportion 80%/20% (in volume), respectively, as determined from the 3-phase Watershed segmentation presented in section 3.2.2 of this manuscript." Grain size was determined by micro CT-image analysis and is between $16\mu$m - $794\mu$m (median $140\mu$m) for quartz grains and $19\mu$m - $446\mu$m (median $168\mu$m) for carbonates grains and between $15\mu$m - $606\mu$m (median $159\mu$m) for quartz and $15\mu$m - $415\mu$m (median $172\mu$m) for carbonate grains for Scarborough and Cottesloe beaches, respectively.

Referee Comment #3 Line 130: how are you sure that after compaction the sandpack is homogeneous?

Authors Reply We do not make any statement in the text as whether the sand pack is homogeneous or not, but simply claim that our experimental method of deposition reproduces a packing as close as possible as the one in-situ.

Authors Changes none

Referee Comment #4 Line 158: you should add some words about the "non-conventional" rectangular cell. Why did you use such a geometry? What was the objective of using this configuration?

Authors Reply Thank you for your comment. Firstly, we will not agree that the rectangular cell is a "non-conventional" one. In the text we have explained the difference between two cells in the operation procedure. However you are absolutely correct that we have to explain why we are using such different geometries.

Authors Changes: After line 156 we have added: Thus, the utilization of this rectangular shape "static cell" drastically reduces the experimental time, moreover the sample

preparation for "static cell" is easier than for "flow cell"

Referee Comment #5 Figures 1 and 2: add the scale

Authors Reply we completed the caption of this figures as follow

Authors Changes The height of the flow cell is 27cm. for Figure 1 and The length of the static cell is 29.8cm. for Figure 2

Referee Comment #6 Line 195, equation 3: use sigma_w instead of C. C is generally used to denote the concentration, not the electrical conductivity.

Authors Reply Yes , I agree with the comment. And this has been corrected in the manuscript.

Authors Changes We have replaced Cw by $\sigma$w

Referee Comment #7 Line 201: maybe show an example of sigma_rock vs sigma_water with the fitting straight line.

Authors Reply This is a very good suggestion and we added the following sentence in the text and an additional figure.

Authors Changes L.201 Added: "Such as a plot is given in Figure 3 for the example of Cottesloe Beach sample with porosity 33%"

Referee Comment #8 Table 1: maybe provide the adjustment coefficient to provide an estimation of the quality of the value of FF

Authors Reply Thanks for the suggestion: however adding the correlation coefficients for all FF would make the tables (already quite large) very difficult to read, so instead we completed the text with the range of Rˆ2 we obtained.

Authors Changes L. 341: we added : "Correlations coefficients were very good to excellent and varied between 0.975 and 0.999 and between 0.974 and 0.996 for the flow cell, for Scarborough and Cottesloe samples, respectively, and between 0.882 and

0.993 and between 0.987 and 0.999 for the static cell, for Scarborough and Cottesloe samples, respectively.

Referee Comment #9 Figure 7: add the unit for the electrical field.

Authors Reply Thanks for the comment. The unit of the electric field here is in order of magnitude of ($\mu$V). This is not the point; the potential electrical field is relative field. The gradient of the electric field is essential for electrical conductivity in the porous media. Which the contrast of colour shows the local change of electric fields, in the near to grain contacts and pore throats this electric field is changing more than inside of pore volume. This also could vary by adding surface conductivity to the grains or clay conductivity in the sample. Our aim for showing these images here is to show the heterogeneity of the potential field, calculated from simulations.

Authors Changes We added the following in the caption of Figure 7 (now 8) "Colorbar indicates regions of high (red) and low (blue) potential field in arbitrary unit"

Referee Comment #10 Figure 8: if I am right, this figure is not referenced in the text. Colorbar and unit are missing.

Authors Reply Please see reply above

Authors Changes we added in the caption: "Color indicates regions of high (red) and low (blue) potential field in arbitrary unit."

Referee Comment #11 Figure 9: could you add the error bars, for both porosity and formation factor? Also add the value of the cementation exponent for the dashed lines corresponding to the fit of the experimental data.

Authors Reply Thank you for your suggestion

Authors Changes We have changed Figure 9 (now 10)

Referee Comment #12 Line 380: to validate your approach, a figure is missing, showing the comparison of the measured and compared value. I suggest you to plot measured FF/porosity and computed FF/porosity, as well as the 1:1 line.

Authors Reply Actually, the point of the method we show here is to compare trends between two properties (e.g FF and porosity) obtained by the two different approaches (lab abd computation), and NOT to compare values. We have explained it in the introduction l. 97 to L.104. Data from the lab and from the computation have been obtained at different scales so they fundamentally DO NOT have to match However we added an additional figure in the discussion that compares laboratory and computation data

Authors Changes An additional Figure (#15) has been added

Referee Comment #13 Discussion: again, a comment on the interest of the unconventional cell is required. A comment about the deviating trend of the measured data for the Cottesloe sand with unconventional cell is missing.

Authors Reply Thank for that comment, we have reflected it in the text

Authors Changes We have added after line 346 the following "Some deviations between the results obtained for both static and flow cells may be due to non-uniform compaction of the samples in a case of the flow cell and or non-complete fluid replacement in the case of flow cell. "

Referee Comment #14 Figure 13: informations are missing in the caption. Which data are from experimentally measured values, from image-computation? The dots corresponding to this study are missing (for comparison). Moreover, the references of the data should be provided (for instance, "from Smith et al.").

Authors Reply: we have completed the caption

Additional Comment Please also refer to the new version of the discussion

[Figure]

[Figure]

**Fig. 1.**

**Fig. 2.**

---

## Author Comment (AC5) · 21 May 2019

Referee Comment: difference between FF of rectangular and cylindrical cells,

Referee Comment: different trends between FF and porosity in Figure 9. For example, there are obviously another trends than FF=PhiЁĘ(-m) in some cases (e.g., Scarborough Beach, flow or Cottesloe Beach, static), but authors insisted on Archie trend. These differences go back to the porous structure of each sample and since the authors have access to it (I mean via segmented images), they should do more extensive work in this part. They can at least calculate pore network model or tortuosity of flow paths of each sample and compare their properties to find reasonable relationships.

[Figure]

Authors Reply: Please refer now to the discussion that has been highly modified and replies to this comment, especially 2nd paragraph.

Please also note the supplement to this comment:
https://www.solid-earth-discuss.net/se-2018-133/se-2018-133-AC5-supplement.pdf

**Supplement:**

**5. Discussion**

As noticed earlier in section 4.1, the values of formation factor obtained by the static cell are higher than that obtained by the dynamic cell (for a given porosity), for both samples. This translates in a higher cementation exponent $m$. One reason for this can be the design of the cell itself and of the way to achieve a stable reading of sample conductivity, for each fluid salinity. In the rectangular (static) cell, because the higher salinity brine is introduced or retrieved via the center of the panels (see Figure 2) there could some brine left in the corners that will only equilibrate with the new injected brine by diffusion and hence there could be a lower conductivity of the brine in these corners compared to the conductivity of the injected brine. As result the measured sample conductivity will be lowered with respect with what it should be, giving a higher ratio sample to brine conductivities (i.e. formation factor, see Eqn. 11). Using a cylindrical cell has thus the advantage of providing a better replacement of the brine.

In Figure 14 are reported data from both literature and those acquired in this study for Cottesloe and Scarborough beach samples (using the flow cell). Data from literature include natural sand samples and synthetic granular media made of plastic particles of regular geometrical shape (Wyllie and Gregory, 1953). We have bounded these data by the relationship presented in Eqn. 14, with $m$=1.3, which corresponds to the original work of Archie (1952) for unconsolidated media and by the same relationship, with $m$=1.8, for the upper bound. We see in this figure that our experimental results for Cottesloe and Scarborough beach samples are in agreement with data reported for other beach sands. Considering the data reported in this figure, we observe that Archie's classical formula for unconsolidated media underestimates the formation factor and that the departure from sphericity leads to a larger $m$ coefficient. Since Archie's work, many authors have proposed alternative formation factor-porosity relationships. Winsauer et al. (1950) suggested that $a \neq 1$ in Eqn. 14 is a better expression, whereas other authors derived non-power laws dependency to porosity. From a practical point of view, no formula relating the formation factor to porosity for unconsolidated media fits all the experimental data, and, for a given porosity, the formation factor depends on the particle geometry, particle size distribution and subsequent packing.

[Figure]

Figure14: Comparison of laboratory results with results from other workers (Wyllie and Gregory, 1953). CB stands for Cottesloe Beach samples and SCB Scarborough Beach samples.

In Figure 15, we compare laboratory data to computed data. Laboratory data are those acquired with the flow cell, which, as discussed earlier in this section, are expected to give more reliable data. Computed data are those obtained for a cube size of $(700)^3$, which is above the REV, as presented in section 4.2. We can see that there is an excellent agreement for Cottesloe beach sample, and a good agreement for Scarborough beach sample. At this stage, it is difficult to explain why one sample gave better agreement, and whether it is due to an experimental error or due to the higher content of carbonate grains for Scarborough sample that make the computation less accurate: indeed carbonate grains may present some intra-porosity (as for example micritic phases) and thus have an electrical conductivity.

[Figure]

Figure 15: Comparison between laboratory results (in open symbols) end computed ones (in plain symbols). The trends in dashed lines are obtained from the laboratory-measured data.